# Bromotryptamine and Bromotyramine Derivatives from the Tropical Southwestern Pacific Sponge *Narrabeena nigra*

**DOI:** 10.3390/md17060319

**Published:** 2019-05-30

**Authors:** Maria Miguel-Gordo, Sandra Gegunde, Kevin Calabro, Laurence K. Jennings, Amparo Alfonso, Grégory Genta-Jouve, Jean Vacelet, Luis M. Botana, Olivier P. Thomas

**Affiliations:** 1Marine Biodiscovery, School of Chemistry and Ryan Institute, National University of Ireland Galway (NUI Galway), University Road, H91 TK33 Galway, Ireland; m.miguelgordo1@nuigalway.ie (M.M.-G.); kevin.calabro@nuigalway.ie (K.C.); laurence.jennings@nuigalway.ie (L.K.J.); 2Departamento de Farmacología, Facultad de Veterinaria, Universidade de Santiago de Compostela, 27002 Lugo, Spain; sandra.gegunde@rai.usc.es (S.G.); amparo.alfonso@usc.es (A.A.); 3Laboratoire de Chimie-Toxicologie Analytique et Cellulaire (C-TAC) UMR CNRS 8038 CiTCoM Université Paris-Descartes, 4, avenue de l’Observatoire, 75006 Paris, France; gregory.genta-jouve@parisdescartes.fr; 4Muséum National d’Histoire Naturelle, Unité Molécules de Communication et Adaptation des Micro-organismes (UMR 7245), Sorbonne Universités, CNRS, 75005 Paris, France; 5Aix Marseille Université, CNRS, IRD, IMBE UMR 7263, Avignon Université, Institut Méditerranéen de Biodiversité et d’Ecologie marine et continentale, Station Marine d’Endoume, Chemin de la Batterie des Lions, 13007 Marseille, France; jean.vacelet@imbe.fr

**Keywords:** Futuna, Porifera, *Narrabeena*, coral reefs, aromatic alkaloids, Bromotryptamine, Bromotyramine, neuroprotective agents

## Abstract

So far, the Futuna Islands located in the Central Indo-Pacific Ocean have not been inventoried for their diversity in marine sponges and associated chemical diversity. As part of the Tara Pacific expedition, the first chemical investigation of the sponge *Narrabeena nigra* collected around the Futuna Islands yielded 18 brominated alkaloids: seven new bromotryptamine derivatives **1**–**7** and one new bromotyramine derivative **8** together with 10 known metabolites of both families **9**–**18**. Their structures were deduced from extensive analyses of nuclear magnetic resonance (NMR) and high-resolution mass spectrometry (HRMS) data. *In silico* metabolite anticipation using the online tool MetWork revealed the presence of a key and minor biosynthetic intermediates. These 18 compounds showed almost no cytotoxic effect up to 10 µM on human neuroblastoma SH-SY5Y and microglia BV2 cells, and some of them exhibited an interesting neuroprotective activity by reducing oxidative damage.

## 1. Introduction

Bioprospection represents the first step of the marine biodiscovery process, followed by the description of bioactive molecules, which can find applications especially in human health and the pharmaceutical sector. The Indo-Pacific Ocean is considered a key geographical area for marine biodiscovery, not only because of its luxuriant marine biodiversity, but also because marine invertebrates of remote islands usually present a high rate of endemicity, then leading to a potential chemical novelty. Importantly, inventories of the marine biodiversity around isolated islands also contribute to a global understanding of our oceans using an integrative approach composed of taxonomy, chemistry, ecology, biology, biochemistry, and microbiology [1,2,3]. The Tara Pacific expedition (2016–2018) explored the Pacific Ocean with the main scientific objective to conduct a comprehensive description of the marine biodiversity present in endangered coral reefs, from genes to ecosystem. In this context, an inventory of marine sponges was achieved in some locations for the first time, with the second objective to describe their associated chemical diversity [4]. The islands of Wallis and Futuna are located in the biodiversity-rich Tropical Southwestern Pacific, and only little information has been given about the biodiversity of marine invertebrates in this territory, mostly around the island of Wallis [5,6]. To the best of our knowledge, no detailed inventory of littoral sponges has been reported so far from the Futuna Islands composed of the two islands: Futuna and Alofi [7,8,9].

Following preliminary chemical profiling of the fractions obtained from sponges collected in this area, the sponge *Narrabeena nigra* Kim and Sim, 2010, first described in Korea [10] was selected for a thorough chemical investigation due to the presence of a high diversity of brominated alkaloids. Even though no natural products have been reported for species of this genus, other species of the Thorectidae family such as *Smenospongia* sp. and *Hyrtios* sp. are known to produce metabolites of this family with cytotoxic [11,12,13], anti-inflammatory, antioxidant [14], and antidepressant activities [15]. We describe herein the isolation and structure elucidation of seven new bromotryptamine metabolites **1**–**7** and a new bromotyramine derivative **8** (Figure 1), along with 10 known brominated analogues: 5,6-dibromo-*N*,*N*-dimethyltryptamine (**9**) [16,17], 5,6-dibromo-*N-*methyltryptamine (**10**) [18], 5,6-dibromotryptamine (**11**) [18], 6-bromo-*N*-methyltryptamine (**12**) [19], 6-bromotryptamine (**13**) [20,21], 6-bromokynuramine (**14**) [22], 7-bromoquinolin-4(1*H*)-one (**15**) [22], 3,5-dibromo-4-methoxytyramine (**16**) [23], 3-bromo-4-methoxy-*N*,*N*,*N*-trimethyltyrosine (**17**) [24], and 3-bromo-4-methoxytyramine (**18**), isolated for the first time as a natural product but widely used as a reactant in synthesis [23]. The biological activity of these alkaloids was assessed on two cellular models associated with neuroinflammation [25]. 

## 2. Results and Discussion

### 2.1. Isolation and Structure Elucidation

The freeze-dried sponge sample (66.8 g) was extracted with MeOH/CH_2_Cl_2_ (1:1) under sonication three times. The extract was then fractionated by reversed phase (RP)-C18 vacuum liquid chromatography (VLC) with solvents of decreasing polarity: H_2_O_,_ H_2_O/MeOH (1:1), MeOH, MeOH/CH_2_Cl_2_ (1:1), and CH_2_Cl_2_. Compounds of the H_2_O/MeOH, MeOH, and MeOH/CH_2_Cl_2_ fractions were then purified by repeated preparative, semi-preparative, and analytical RP-HPLC, leading to the isolation of 18 pure metabolites, including the new tryptamine alkaloids **1**–**7** and the new tyramine derivative **8**. As the known 5,6-dibromo-*N*,*N*-dimethyltryptamine (**9**) and 3,5-dibromo-4-methoxytyramine (**16**) were found to be the major metabolites of the extract, the structures of the new metabolites were mainly deduced by comparison of their nuclear magnetic resonance (NMR) and high resolution mass spectrometry (HRMS) spectra with those of both **9** and **16**. 

Compound **1** was isolated as a yellow amorphous solid, and its HRMS spectrum revealed the molecular formula C_13_H_16_Br_2_ClN_2_ calculated from the ion at *m/z* 392.9366 [M]^+^ and an isotopic pattern characteristic of the presence of two bromine and one chlorine atoms. The ^1^H NMR spectrum of **1** evidenced the presence of a 3, 5, 6-trisubstituted indole ring system with characteristic singlets at δ_H_ 7.30 (s, H-2), 7.98 (s, H-4), and 7.74 (s, H-7) (Table 1). Two additional signals of the AA’XX’ system at δ_H_ 3.27 (H-8) and the deshielded signal at δ_H_ 3.73 (H-9) were reminiscent of a quaternary tryptammonium ion. The signal at δ_H_ 3.33 (s, H_3_-11 and H_3_-11′) first revealed two equivalent methyls placed on the terminal amine. When comparing with known derivatives of this family, the NMR data of **1** were similar to those of the known 5,6-dibromo-*N*,*N*-dimethyltryptamine (**9**), with the presence of an additional deshielded signal of a methylene group at δ_H_ 5.39 (s, H_2_-12) and δ_C_ 68.2 (C-12) (Table 1 and Table 2) [17]. These unusual signals perfectly matched with the signals corresponding to the rare *N*-(chloromethyl) substituent, as exemplified by NMR data of plant *N*-(chloromethyl)tryptamine derivatives [26]. The structure was confirmed first using the key H-12/C-9, C-11 and C-11′ heteronuclear multiple bond correlation (HMBC) correlations and then the fragments at *m/z* 344.9466 and 301.8995 in the HRMS/MS spectrum of **1**, indicative of the loss of a *N*-chloromethyl and chloromethyldimethylamine moieties, respectively. Therefore, **1** was identified as the trifluoroacetate (TFA) salt of 5,6-dibromo-*N*-chloromethyl-*N*,*N*-dimethyltryptammonium. *N*-chloromethyl derivatives have been commonly found as artefact products of the alkylation of tertiary amines by dichloromethane [27]. As we were unable to find a trace of **1** by ultra-high performance liquid chromatography (UHPLC)-HRMS/MS analysis of the ethanolic extract prepared from the same sponge specimen, we could conclude that **1** is produced during the extraction process. 

The molecular formula of **2**, isolated as a colorless amorphous solid, was deduced as C_12_H_14_Br_2_N_2_O from the molecular ion at *m/z* 360.9557 [M + H]^+^, showing an isotopic pattern of two bromine atoms. While the ^1^H NMR and ^13^C NMR spectra were very similar to those of **9**, the signals corresponding to the H_2_-9 methylene and the two *N*-methyls were strongly deshielded. Because **2** showed a molecular mass 16 amu higher than **9**, it was proposed to be the *N*-oxide analogue of **9**. This assumption was confirmed by the presence of a fragment at *m/z* 301.9008 in HRMS/MS, revealing the loss of the *N*,*N*-dimethylamine-*N*-oxide fragment. Thus, **2** was established as 5,6-dibromo-*N*,*N*–dimethyltryptamine-*N*-oxide. 

Compound **3** was isolated as a colorless amorphous solid, showing a molecular ion at *m/z* 342.9433 [M + H]^+^, corresponding to the molecular formula C_12_H_12_Br_2_N_2_. The aromatic region of the ^1^H NMR spectrum was consistent with a 5,6-dibromosubstituted indole. However, the signal corresponding to H-2 was missing when compared to **1** and **2**, indicating this position was substituted. While only one nitrogenated methyl was evidenced in **3** at δ_H_ 3.12 (s, H_3_-11), other broad signals appeared in the ^1^H NMR at δ_H_ 4.62, 4.45 (H_2_-11′), 3.83, 3.53 (H_2_-9) in addition to the signal of the methylene protons at δ_H_ 3.11 (t, *J* = 6 Hz, H_2_-8). The correlation spectroscopy (COSY) and heteronuclear single quantum coherence (HSQC) spectra were not helpful to solve the structure due to the broadening of the signals of the two first methylene groups. Gratifyingly, key H-11/C-11′ and C-9 HMBC correlations were highly informative to place the two methylene groups next to the tertiary amine. The only possibility to comply with the molecular formula and the substitution at C-2 was therefore to envisage the presence of a *N*-methyl substituted tetrahydro-*β*-carboline. Broadening of the signals at C-9 and C-11′ is easily explained as the protonated tertiary amine becomes chiral in the acidic medium used during the purification process [28]. This is the first report of the 6,7-dibromo-2-methyltetrahydro-*β*-carboline.

Compound **4** was obtained as a yellow amorphous solid, and its molecular formula C_12_H_14_Br_2_N_2_O_2_ was deduced from the protonated adduct at *m/z* 376.9503 [M + H]^+^ in its HRMS spectrum. The ^1^H NMR spectrum again revealed the presence of a 5,6-dibromosubstituted indole ring system but, as in **3**, the signal corresponding to H-2 was absent in **4**. The ^13^C NMR spectrum revealed the presence of two new non-protonated carbons: A carbonyl group at δ_C_ 180.3 (qC, C-2) and an oxygenated sp^3^ carbon at δ_C_ 75.3 (qC, C-3). A key H-4/C-3 HMBC correlation placed the non-protonated oxygenated carbon at C-3, while a unique H-8a/C-2 HMBC correlation was consistent with the carbonyl group at C-2. The C-2/C-3 bond of the indole ring of **4** was therefore oxidized into a 3-hydroxyindolin-2-one, and the chemical shifts of the carbons at both positions were in accordance with those of analogues in this series [29]. Two non-equivalent methylenes coupled in the ABMX system at δ_H_ 2.43 (dt, *J* = 15.0, 7.5 Hz, H-8a), 2.06 (ddd, *J* = 15.0, 7.5, 5.0 Hz, H-8b), and 3.58 (dt, *J* = 15.0, 7.5 Hz, H-9a), 3.37 (m, H-9b), therefore confirming the presence of a chiral center at C-3. The absolute configuration at C-3 was assessed by comparison between the experimental and calculated electronic circular dichroism (ECD) spectra. The ECD spectra of both enantiomers of **4** were calculated using time-dependent density functional theory (TDDFT) at the B3LYP/6-311+G(d,p)//B3LYP/6-31G(d) level of theory. Not surprisingly, the calculated ECD spectrum of the 3*R* enantiomer matched the experimental spectrum of **4**, as this configuration is also found for most of the natural products containing a 3-hydroxyindolin-2-one (Figure 2). Finally, **4** was named narrabeenamine A.

Compound **5** was isolated as a yellow amorphous solid, and its molecular formula C_12_H_15_Br_2_N_2_O was deduced from the ion peak of the HRMS spectrum at *m/z* 360.9551 [M]^+^. Like **3** and **4**, the 5,6-dibromosubstituted benzene ring system was deduced from the two aromatic proton singlets at δ_H_ 7.62 (s, H-4) and 7.14 (s, H-7), but the signal corresponding to H-2 was again absent. The presence of a non-protonated oxygenated carbon signal at δ_C_ 88.4 (qC, C-3), together with the key H-4/C-3 HMBC correlation, suggested the presence of a hydroxyl group at C-3. Since no carbonyl signal was observed in the ^13^C NMR spectrum, a different substitution pattern at C-2 was deduced for **5** when compared to **4**. The ^1^H and ^13^C NMR spectra of **5** revealed the presence of one methine group with the proton and carbon signals at δ_H_ 5.18 (s, H-2) and δ_C_ 100.7 (CH, C-2) and two non-equivalent *N*-methyl groups at δ_H_ 3.25 (s, H_3_-11) and 3.01 (s, H_3_-11′). Moreover, H-2/C3, C-7a, C-9, and H-11/C-2, C-9 HMBC correlations revealed the presence of a third ring system containing the aminal functional group of a hexahydropyrrolo[2,3-*b*]indole skeleton (Figure 3). Comparison of the NMR data of **5** with those of other natural products in this series confirmed our assumption and **5** was named narrabeenamine B [30].

The absolute configurations at the C-2 and C-3 chiral centers were assessed by comparison between the experimental and calculated ECD spectra of the 4 possible diastereoisomers. Indeed, the NOESY spectrum did not allow the assignment of the relative configurations even if the spectrum was run in DMSO-*d*_6_. The ECD spectra of the four possible configurations of **5** were therefore calculated using TDDFT at the B3LYP/6-311+G(d,p)//B3LYP/6-31G(d) level of theory. As in **4**, the negative Cotton effect at 250 nm suggested a 3*R* configuration. Comparison between the calculated ECD spectra of both (2*S*, 3*R*) and (2*R*, 3*R*) epimers and the experimental ECD spectrum of **5** evidenced the presence of an additional key Cotton effect of a π→π* transition at approximately 290 nm, which is associated with the configuration at C-2. The negative Cotton effect observed at 310 nm in the experimental spectrum of **5** was in accordance with the (2*S*, 3*R*) relative configurations (Figure 4).

The molecular formula C_9_H_10_Br_2_N_2_O of **6**, a yellow amorphous solid, was deduced from the ion at *m/z* 320.9239 [M + H]^+^ of its HRMS spectrum. The ^1^H NMR spectrum of **6** exhibited two aromatic proton signals at δ_H_ 7.98 (s, H-4) and 7.19 (s, H-7), suggesting again the presence of the 5,6-dibromobenzene ring of the indole ring system and two coupled methylene protons. However, the signals corresponding to H-2 and the methyls were absent, while the presence of a ketone was evidenced by the signal at δ_C_ 198.9 (qC, C-3) in the ^13^C NMR spectrum. The key H-4/C3, C-3a, C-5, C-6, C-7a HMBC correlations placed the ketone at the *ortho* position of the aromatic amino group (C-3 respecting the previous numbering), and the ethylene moiety connected the ketone to another primary amine. Cleavage of the C-2/C-3 bond and loss of the resulting formamide were therefore proposed in order to match the molecular formula. This skeleton is also found in natural products, as exemplified in 5,6-dibromokynuramine [22]. 

Compound **7** was obtained as a yellow amorphous solid and showed a main ion at *m/z* 350.9344 [M + H]^+^ in its HRMS spectrum, leading to the molecular formula C_10_H_12_Br_2_N_2_O_2_. The ^1^H NMR and HSQC spectra revealed the presence of one aromatic proton signal at δ_H_ 7.87 (s, H-4) and a signal of a methoxy group at δ_H_ 3.81 (s, CH_3_O−), while the other signals were very similar to those of **6**. The location of the methoxy group at C-7 of the aromatic ring was inferred from the key CH_3_-O/C-7 HMBC correlation. Therefore, **7** is the methoxylated analogue of **6** at C-7.

Compound **8** was isolated as a colorless amorphous solid with a molecular formula C_10_H_13_Br_2_NO as deduced from the [M + H]^+^ ion at *m/z* 321.9447. The ^1^H NMR spectrum of **8** did not correspond to an indole derivative, as the only aromatic signal at δ_H_ 7.55 (s, H_2_-2/6) was integrated for two protons. The symmetry for the aromatic ring of a dibrominated compound quickly led us to propose a tyramine derivative for **8**. The analogy with the known 3,5-dibromo-4-methoxytyramine (**16**) was evident, and the additional nitrogenated methyl observed at δ_H_ 2.72 (s, CH_3_NH−) revealed that **8** is indeed the new *N*-methyl analogue of **16**.

### 2.2. Biological Assays

Since human neuroblastoma SH-SY5Y and microglia BV2 cells are commonly used for biological studies of neuroinflammation and neuroprotection [31,32], all isolated brominated alkaloids were then tested in these two cellular models. First, the effects of these compounds on cell viability were determined using the 3-(4,5-dimethylthiazol-2-yl)-2,5-diphenyltetrazolium bromide (MTT) assay. Cells were treated with different concentrations of compounds (0.001, 0.01, 0.1, 1, and 10 μM) for 24 h. None of the tested compounds induced cytotoxic effects on BV2 cells at any concentrations tested. Furthermore, among the 18 compounds tested on SH-SY5Y neuroblastoma cells, only **3** at 10 μM reduced cell viability up to 60% (*p* < 0.05) versus control cells. To evaluate the neuroprotective effects of these compounds, *tert*-butyl hydroperoxide (TBHP) was used to induce oxidative damage, and the antioxidant vitamin E was used as a control for neuroprotective effects. As shown in Figure 5, the oxidative damage induced after 6 h of treatment in the presence of TBHP significantly reduced SH-SY5Y cell viability (50%) compared with the control group (*p* < 0.001). This effect was reduced in the presence of vitamin E, restoring cell survival to 80% (*p* < 0.05). In a similar way, some of the brominated alkaloids protected SH-SY5Y cells against TBHP-induced oxidative damage, avoiding cell death. Compound **5** reduced cellular death at the same level as vitamin E and therefore showed a protective effect at 0.01 and 0.1 μM (*p* < 0.05, Figure 5). Furthermore, the same effect was observed in the presence of **7** at all concentrations tested (*p* < 0.05, Figure 5). Finally, the most potent activity was observed after treatment with **15**, since neuronal death induced by TBHP was almost totally inhibited, having a stronger neuroprotective effect than vitamin E (*p* < 0.05, Figure 6). Compounds **9**, **11**, **12**, and **18** prevented cell death in a dose-dependent manner, being statistically significant at the highest concentrations tested (*p* < 0.05, Figure 6). Finally, **10** and **13** reduced TBHP-induced cell death at 0.1 and 1 μM (*p* < 0.05, Figure 5). The rest of the tested compounds did not show any protective effect in SH-SY5Y cells (Figure 5).

To better assess the potential of the nine neuroprotective compounds, their anti-inflammatory activity was tested in activated microglia cells. The activation of microglia leads to the release of pro-inflammatory mediators such as nitric oxide (NO) and the overproduction of these mediators cause oxidative damage in neurons [33]. Therefore, to simulate inflammatory conditions, microglia BV2 cells were activated with lipopolysaccharide (LPS). As shown in Figure 6, when cells were treated with LPS, NO release was doubled compared to control cells (*p* < 0.001). Nevertheless, in the presence of **11** and **15** (0.1 and 1 µM) or **18** (1 µM), the NO release was significantly inhibited (*p* < 0.01). Surprisingly, this effect was also observed after incubation in the presence of 0.1 µM of **9**, **10**, **12**, and **13** but not at 1 µM. In the same conditions, **5** and **7** did not show any significant activity on the microglia BV2 cells. In both the BV2 and SH-SY5Y cellular models, some brominated alkaloids of this family possess interesting properties in neuroinflammation and neuroprotection.

### 2.3. Biosynthetic Considerations

Both bromotryptamine and bromotyramine families of alkaloids found in the Pacific sponge *Narrabeena nigra* mirror the high potential for marine biodiscovery of sponges found around the Futuna Islands. A microbial origin is likely to be involved in the biosynthesis of these compounds, as analogues of these simple brominated aromatic alkaloids were also found in other groups of invertebrates. Among the five different classes of biogenic metabolites of the bromotryptophan family, the oxindole moiety was indeed reported in the convolutamydines, and the pyrroloindole skeleton in the flustramines, both classes of compounds isolated from bryozoans [34,35,36,37,38]. The *β*-carboline moiety has already been found in bryozoans [39] and marine sponges from the genus *Hyrtios* [40]. The quinolone **15** was previously described in the sponge *Clathria basilana* and in a bryozoan [41,42], while the kynuramines, as exemplified by **6**, were isolated in an undescribed sponge from the Red Sea [22]. Additionally, the known tryptamine derivatives **9–13** have been found in marine invertebrates such as gorgonians [19], tunicates [20], and different sponges of the genera *Ancorina* [43], *Geodia* [21], *Jaspis* [44], *Verongula* [15], *Hyrtios* [14,45], *Aplysina* [46], but mostly in *Smenospongia* [16,17,18]. The known tyramines **16**–**18** have been already described in verongiid sponges, [24,47,48] and in an ascidian [23]. Even though both families of compounds were found in *N. nigra*, the majority are bromotryptophan metabolites that are chemotaxonomically related to Dictyoceratida sponges, whereas the bromotyrosine derivatives are associated with the order Verongiida. 

Due to the outstanding diversity of bromotryptamine derivatives isolated in this sponge, we postulate simple interconnections among all the isolated metabolites through metabolic transformations that include methylation and different types of oxidation of the indole nucleus (Scheme 1). 

To further expand the chemical diversity in this family, the web server Metwork [49] performing an in silico metabolite anticipation was used to assess the presence of minor metabolites involved in this metabolic pathway. As expected, some minor metabolites were identified after comparison of the experimental with calculated MS/MS spectra. These compounds (in orange in Figure 7), which are slightly different from the isolated compounds, could correspond to some biosynthetic intermediates. This is especially true for the formylated compound at *m/z* 271.0076 resulting from the oxidative cleavage of 6-bromotryptamine (**14**), which was recently elucidated biosynthetically [50]. Gratifyingly, the cosines of most of the proposed structures are all above 0.5, therefore expressing a high level of confidence (see Appendix A) [51]. The proposed structures are also in perfect agreement with the metabolome consistency, using simple and already confirmed biosynthetic transformations [52].

## 3. Materials and Methods 

### 3.1. General Experimental Procedures

Optical rotation measurements were performed at the Na D line (589.3 nm) with a 5 cm cell at 20 °C on a UniPol L1000 polarimeter (Schmidt + Haensch, Berlin, Germany). UV and ECD data were obtained on ChirascanTM V100 (Applied Photophysics, Leatherhead, UK). NMR experiments were performed on an Inova 500 MHz spectrometer (Varian, Palo Alto, CA, USA) and on a 600 MHz spectrometer (Agilent, Santa Clara, CA, USA). Chemical shifts were referenced in ppm to the residual solvent signals (CD_3_OD, at δ_H_ 3.31 and δ_C_ 49.00 ppm; DMSO-*d*_6_, at δ_H_ 2.49 and δ_C_ 39.5 ppm). High-resolution mass spectra were obtained with a mass spectrometer UHPLC-HRMS (Agilent 6540, Santa Clara, CA, USA). Purifications were performed using several HPLC-DAD: Jasco (Tokyo, Japan) equipped with PU-2087 pump and UV-2075 detector (preparative), Waters 2690 (Milford, MA, USA) equipped with UV detector 2487 (semipreparative and analytical) and Agilent 1260 (Santa Clara, CA, USA) (analytical).

### 3.2. Animal Material.

The specimen of *Narrabeena nigra* Kim and Sim 2010 was collected at 8 m depth around the Alofi Island coast (14°20′30″ S, 178°04′53″ W), in December 2016, during the Tara Pacific expedition. A fragment was fixed with EtOH for taxonomic studies, while the rest of the sample was frozen at –80 °C and freeze-dried for chemical studies. Voucher specimen n° 161213Fu06-01 is stored at NUIG (National University of Ireland, Galway, Ireland).

### 3.3. Extraction and Purification

The lyophilized and ground sponge (66.8 g) was extracted with MeOH/CH_2_Cl_2_ at room temperature under sonication (1:1; 3 × 400 mL, 5 min), and the solution was evaporated under reduced pressure. The dried extract (7.27 g) was sequentially fractionated in five fractions by flash silica C-18 VLC with solvents of decreasing polarity: (1) H_2_O; (2) H_2_O/MeOH (1:1); (3) MeOH; (4) MeOH/CH_2_Cl_2_ (1:1); and (5) CH_2_Cl_2_. 

Fraction 2 (2.36 g) was purified by repeated semi-preparative reversed phase (RP)-HPLC (Waters SymmetryPrep C18, 7 µm; 7.8 × 300 mm; flow rate: 3.5 mL/min; UV detection: 210 nm), using a gradient of solvents H_2_O:CH_3_CN/0.1% TFA (80:20, 5 min; ramp to 70:30 over 20 min; 70:30 for 5 min), which led to 10 peaks (F2P1-F2P10), including **4** (*t*_R_ = 15.8 min, 1.95 mg, 2.92 × 10^−5^% w/w), **9** (*t*_R_ = 24.0 min, 72.8 mg, 1.09 × 10^−3^% w/w) and **16** (*t*_R_ = 17.5 min, 30.7 mg, 4.60 × 10^−4^% w/w). Further purification of F2P1 (Waters XSelect HSS T3, 5 µm; 4.6 × 250 mm; flow rate: 1 mL/min; UV detection: 210 nm) with an isocratic solvent composition of H_2_O:CH_3_CN/0.1% TFA (82:18), led to **17** (*t*_R_ = 18.7 min, 1.23 mg, 1.84 × 10^−5^% w/w) and **18** (*t*_R_ = 14.9 min, 1.50 mg, 2.25 × 10^−5^% w/w). The purification of F2P5 (Waters Xselect Phenyl-hexyl, 5 µm; 4.6 × 250 mm; flow rate: 1 mL/min; UV detection: 210 nm) using an isocratic solvent system H_2_O:CH_3_CN/0.1% TFA (86:14) provided **6** (*t*_R_ = 36.5 min, 3.92 mg, 5.87 × 10^−5^% w/w) and **16** (*t*_R_ = 31.5 min, 3.25 mg, 4.87 × 10^−5^% w/w) and the separation of F2P9 (3.87 mg) (Waters Xselect Phenyl-hexyl, 5 µm; 4.6 × 250 mm; flow rate: 1 mL/min; UV detection: 210 nm) with a gradient of solvents H_2_O:CH_3_CN/0.1% TFA (80:20, 5 min; ramp to 50:50 over 30 min) led to **9** (*t*_R_ = 13.8 min, 1.50 mg, 2.25 × 10^−5^% w/w), **10** (*t*_R_ = 13.2 min, 1.32 mg, 1.98 × 10^−5^% w/w) and **11** (*t*_R_ = 12.5 min, 0.91 mg, 1.36 × 10^−5^% w/w). 

Fraction 3 (0.38 g) was purified using the same conditions as those used for fraction 2, and 10 peaks were obtained (F3P1-F3P10), including **2** (*t*_R_ = 29.8 min, 1.99 mg, 2.98 × 10^−5^% w/w), **9** (*t*_R_ = 22.5 min, 37.2 mg, 5.57 × 10^−4^% w/w), and **18** (*t*_R_ = 9.2 min, 2.48 mg, 3.71 × 10^−5^% w/w). Then, the purification of F3P2 (Waters Xselect Phenyl-hexyl, 5 µm; 4.6 × 250 mm; flow rate: 1 mL/min; UV detection: 210 nm) was done using an isocratic system of H_2_O:CH_3_CN/0.1% TFA (87:13) led to **14** (*t*_R_ = 14.4 min, 0.69 mg, 1.03 × 10^−5^% w/w) and **15** (*t*_R_ = 22.9 min, 1.06 mg, 1.59 × 10^−5^% w/w). 

The combined F2P4 and F3P4 (6.79 mg) was further purified (Waters Xselect Phenyl-hexyl, 5 µm; 4.6 × 250 mm; flow rate: 1 mL/min; UV detection: 210 nm) with H_2_O:CH_3_CN/0.1% TFA (86:14), which gave **5** (*t*_R_ = 34.0 min, 1.74 mg, 2.61 × 10^−5^% w/w), **12** (*t*_R_ = 31.5 min, 1.20 mg, 1.80 × 10^−5^% w/w), and **13** (*t*_R_ = 26.5 min, 1.32 mg, 1.98 × 10^−5^% w/w). The combined F2P7 and F3P7 (6.02 mg) was further purified (Waters Xselect Phenyl-hexyl, 5 µm; 4.6 × 250 mm; flow rate: 1 mL/min; UV detection: 210 nm) with H_2_O:CH_3_CN/0.1% TFA (82:18), affording **7** (*t*_R_ = 23.9 min, 2.66 mg, 3.98 × 10^−5^% w/w), **8** (*t*_R_ = 19.9 min, 1.10 mg, 1.65 × 10^−5^% w/w), and **16** (*t*_R_ = 17.0 min, 0.66 mg, 9.88 × 10^−6^% w/w).

Fraction 4 (1.03 g) was purified by RP-HPLC (Waters Xselect Prep C18, 5 µm; 19 × 250 mm; flow rate: 12 mL/min; UV detection: 210 nm), using an isocratic solvent H_2_O:CH_3_CN/0.1% TFA (70:30). Further purification of combined F4P6 and F4P7 (10.66 mg) (Waters Xselect Phenyl-hexyl, 5 µm; 4.6 × 250 mm; flow rate: 1 mL/min; UV detection: 210 nm) using H_2_O:CH_3_CN/0.1% TFA (75:25) led to **1** (*t*_R_ = 18.5 min, 0.92 mg, 1.38 × 10^−5^% w/w), **3** (*t*_R_ = 16.0 min, 2.24 mg, 3.35 × 10^−5^% w/w), **9** (*t*_R_ = 13.0 min, 2.39 mg, 3.58 × 10^−5^% w/w), and **10** (*t*_R_ = 11.8 min, 1.44 mg, 2.16 × 10^−5^% w/w).


*5,6-Dibromo-N-chloromethyl-N,N-dimethyltryptammonium (*
**1**
*)*


Yellow amorphous solid; UV (MeOH) *λ*_max_ 230, 295 nm; ^1^H NMR and ^13^C NMR data, Table 1 and Table 2; ESI(+)-HRMS *m/z* 392.9366 [M]^+^ (calcd. for C_13_H_16_Br_2_ClN_2_, 392.9363, Δ +0.8 ppm). 


*5,6-Dibromo-N,N–dimethyltryptamine-N-oxide (*
**2**
*)*


Colorless amorphous solid; UV (MeOH) *λ*_max_ 230, 295 nm; ^1^H NMR and ^13^C NMR data, Table 1 and Table 2; (+)-HRESIMS *m/z* 360.9557 [M + H]^+^ (calcd. for C_12_H_15_Br_2_N_2_O, 360.9546, Δ +3.0 ppm). 


*6,7-Dibromo-2-methyltetrahydro-β-carboline (*
**3**
*)*


Colorless amorphous solid; UV (MeOH) *λ*_max_ 231, 290 nm; ^1^H NMR and ^13^C NMR data, Table 1 and Table 2; (+)-HRESIM S *m/z* 342.9433 [M + H]^+^ (calcd. for C_12_H_13_Br_2_N_2_, 342.9440, Δ −2.0 ppm). 


*Narrabeenamine A (*
**4**
*)*


Yellow amorphous solid; [α]_D_^20^ +12 (*c* 0.1, MeOH); UV (CH_3_CN) *λmax* (log ɛ) 218 (3.66), 260 (3.23), 310 (2.83) nm; ECD (*c* 2.7 × 10^−4^ M, CH_3_CN) *λ*_max_ (∆*ε*) 220 (+0.34), 246 (−0.32), 274 (+0.1) nm; ^1^H NMR and ^13^C NMR data, Table 1 and Table 2; (+)-HRESIMS *m/z* 376.9503 [M + H]^+^ (calcd. for C_12_H_15_Br_2_N_2_O_2_, 376.9495, Δ +2.1 ppm). 


*Narrabeenamine B (*
**5**
*)*


Yellow amorphous solid; [α]_D_^20^ +20 (*c* 0.1, MeOH); UV (CH_3_CN) *λ*_max_ (log *ɛ*) 246 (3.52), 310 (2.96) nm; ECD (*c* 5.5 × 10^−4^ M, CH_3_CN) *λ*_max_ (∆*ε*) 245 (−0.12), 270 (−0.01), 310 (−0.05) nm; ^1^H NMR and ^13^C NMR data, Table 1 and Table 2; (+)-HRESIMS *m/z* 360.9551 [M]^+^ (calcd. for C_12_H_15_Br_2_N_2_O, 360.9546, Δ +1.1 ppm). 


*5,6-Dibromokynuramine (*
**6**
*)*


Yellow amorphous solid; UV (MeOH) *λ*_max_ 235, 264, 376 nm; ^1^H NMR and ^13^C NMR data, Table 1 and Table 2; (+)-HRESIMS *m/z* 320.9239 [M + H]^+^ (calcd. for C_9_H_11_Br_2_N_2_O, 320.9233 Δ +1.9 ppm). 


*5,6-Dibromo-7-methoxykynuramine (*
**7**
*)*


Yellow amorphous solid; UV (MeOH) *λ*_max_ 240, 265, 377 nm; ^1^H NMR and ^13^C NMR data, Table 1 and Table 2; (+)-HRESIMS *m/z* 350.9344 [M + H]^+^ (calcd. for C_10_H_13_Br_2_N_2_O_2_, 350.9338, Δ +1.7 ppm). 


*3,5-Dibromo-4-methoxy-N-methyltyramine (*
**8**
*)*


Colorless amorphous solid; UV (MeOH) *λ*_max_ 208, 280 nm; ^1^H NMR (500 MHz, CD_3_OD) δ_H_ 7.55 (s, H-2 and H-6), 3.85 (s, H-11), 3.24 (t, *J* = 7.5 Hz, H-8), 2.93 (t, *J* = 7.5 Hz, H-7), 2.72 (s, H-10); ^13^C NMR; (125 MHz, CD_3_OD) δ_C_ 154.7 (C, C-4)), 136.7 (C-1), 134.3 (C-2, C-6), 119.8 (C-3, C-5), 61.1 (C-11), 50.8 (C-8), 33.7 (C-10), 31.8 (C-7); (+)-HRESIMS *m/z* 321.9447 [M + H]^+^ (calcd. for C_10_H_14_Br_2_NO, 321.9437, Δ +3.1 ppm). 

### 3.4. Computational Methods

ECD. The low energy conformers of each compound were generated using the Schrodinger MacroModel 11.3 software package in Maestro release 2017-4 (Schrödinger, LLC, New York, NY, USA) as described previously [53]. The conformers were optimized using Gaussian 16 (Wallingford, CT, USA) at the B3LYP/6-311+G(d,p) level of theory, while at the same time, the zero-point energy, electronic transition, and rational strength of conformers were calculated for the free-energy distribution of the conformers [54]. The ECD spectrum was calculated using Gaussian 16 at the B3LYP/6-31G(d) level, and spectra were produced using the freely available software SpecDis 1.7 (Berlin, Germany) [55]. All calculations were performed using a polarizable continuum model with acetonitrile. The calculated spectra were then compared to the experimental spectra.

MetWork v0.3.5, Paris, France. Raw MS/MS data were converted into .mgf using MSconvert v3.0.18105-622e002cb, and a molecular network was created using the online workflow at GNPS [51]. The data were filtered by removing all MS/MS peaks within +/− 17 Da of the precursor *m/z*. MS/MS spectra were window filtered by choosing only the top 6 peaks in the +/− 50 Da window throughout the spectrum. A network was then created where edges were filtered to have a cosine score above 0.7 and more than 3 matched peaks. Further edges between two nodes were kept in the network if and only if each of the nodes appeared in each other’s respective top 10 most similar nodes. The corresponding clustered .mgf was then uploaded to the MetWork server [49]. Compound **13** was used for the in silico metabolization using indole-related biotransformations. For the comparison between experimental and predicted MS/MS spectra, a cosine value threshold of 0.45 was used.

### 3.5. Bioassays

Cell Culture

Murine microglia BV-2 cell line was purchased from InterLab Cell Line Collection (ICLC, Genoa, Italy), number ATL03001. Cells were maintained in Roswell Park Memorial Institute Medium (RPMI), plus 10% fetal bovine serum (FBS), 100 µg/mL streptomycin and penicillin (100 U/mL) at 37 °C in a humidified atmosphere of 5% CO_2_ and 95% air. Cells were dissociated twice a week using 0.05% trypsin/ethylenediaminetetracetic acid (EDTA). 

Human neuroblastoma SH-SY5Y cell line was obtained from American Type Culture Collection (ATCC), number CRL2266. Cells were maintained in Dulbecco’s Modified Eagle Medium: Nutrient Mix F-12 (DMEN/F-12) plus 10% FBS, 1% glutamax, 100 µg/mL streptomycin and penicillin (100 U/mL) at 37 °C in a humidified atmosphere of 5% CO_2_ and 95% air. Cells were dissociated once a week using 0.05% trypsin/EDTA. 

Cell Viability

The MTT assay was used to analyze cell viability. Briefly, microglia BV2 cells were cultured in 384 well plates at a density of 2 × 10^4^ cells per well or 2.5 × 10^4^ cells per well in the case of neuroblastoma SH-SY5Y cells. Cells were incubated with different compound concentrations (0.001, 0.01, 0.1, 1, and 10 µM) for 24 h. Then, cells were washed and incubated with MTT [3-(4,5-dimethyl thiazol-2-yl)-2,5-diphenyl tetrazolium bromide] (500 µg/mL) diluted in phosphate buffered saline (PBS) for 1 h at 37 °C. The resulting formazan crystals were dissolved with sodium dodecyl sulfate (SDS), and the absorbance was measured on a spectrophotometer plate reader at 595 nm (Bio-Tek Synergy, Winooski, VT, USA). 

Neuroprotection Assay

The neuroprotective effects on cellular viability of compounds in the presence of TBHP were measured by the MTT assay as described above. For this, cells were incubated with compounds at different concentrations (0.001, 0.01, 0.1, 1, and 10 µM) and TBHP (65 µM) for 6 h. The known antioxidant vitamin E (25 µM) was used as a positive control for neuroprotective activity.

NO Determination

The NO concentration in the culture medium was determined using the Griess reagent kit (Thermo Fisher, Madrid, Spain), in accordance with manufacturer’s instructions. Briefly, microglia BV2 cells were seeded in 24-well plates (1 × 10^6^ cells per well) and incubated with compounds (1 and 0.1 µM) 1 h before the stimulation with LPS (500 ng/mL) for 24 h. Next, in a 96-well plate it was added 130 μL of deionized water, 150 μL of cells in culture medium, and 20 μL of Griess reagent and then it was incubated for 30 min in the dark and at room temperature. The absorbance was measured on a spectrophotometer plate reader at 548 nm (Bio-Tek Synergy).

## 4. Conclusions 

This work represents the first chemical study of a sponge from the genus *Narrabeena*. The chemical investigation of the Pacific sponge *Narrabeena nigra* collected around the Futuna Islands led to the isolation of a large diversity of simple bromotryptamine and bromotyramine derivatives. As analogues of these families were also found in other marine invertebrates, we hypothesize a microbial origin for these compounds. The use of the webserver MetWork allowed the identification of minor possible biosynthetic intermediates through natural product anticipation based on comparison with calculated MS/MS data. 

Overproduction of reactive oxygen species (ROS) generates an oxidative stress state which is related to neurodegenerative diseases. Oxidative stress occurs upon an excessive ROS production and deficiency of an antioxidant response [56]. Therefore, compounds able to protect neurons against oxidative damage are excellent candidates to be used in neurodegenerative disorders, such as Alzheimer’s or Parkinson’s disease [57]. In the present work, human neuroblastoma SH-SY5Y cells treated with TBHP for 6 hours were consolidated as a model of oxidative stress damage. Some of the brominated alkaloids, like **5**, showed an interesting neuroprotective effect against TBHP-induced oxidative damage in SH-SY5Y cells, in the same way as the potent antioxidant vitamin E. In addition, seven of these compounds decreased the release of the neurotoxic mediator NO in activated microglia. Activated microglia play a crucial role in neuroinflammation through the excessive production of pro-inflammatory mediators. Further, microglia-mediated inflammation has been related to neurodegenerative diseases [58]. Moreover, these brominated alkaloids showed very low toxicity, up to 10 µM, in neuron and microglia cell lines. All these results suggest the potential of these natural products as a therapeutic tool to prevent neuronal cell death in age-associated diseases. A recent publication confirms the potential of some oxidized analogues in this series [59]. Nevertheless, further studies will be necessary to better understand the mechanism of action of these compounds.

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
