# Peer review of "Bromotryptamine and Bromotyramine Derivatives from the Tropical Southwestern Pacific Sponge Narrabeena nigra"

_marinedrugs, 2019, doi:10.3390/md17060319_

Round 1

Reviewer 1 Report

This paper from Gordo M.M. and colleagues describes the isolation of 18 bromotryptamine and bromotyramine derivatives from the tropical sponge Narrabeena nigra. Eight of these compounds have been described for the first time and their chemical structures have been determined by NMR, HRMS and ECD analysis. An in silico metabolite anticipation has been also performed to assess the presence of minor metabolites involved in the metabolic pathway leading to the bromotryptophan alkaloids isolated in N. nigra.

The 18 purified compounds have been tested to evaluate their neuroprotective effect against TBHP-induced oxidative damage in SH-SY5Y cells. Some of them showed the same antioxidant effect as vitamin E. Moreover, seven of these compounds decreased the release of the neurotoxic mediator NO in activated BV2 microglia cell line. Finally, none of these brominated alkaloids have shown toxicity, up to 10 μM concentration.

The manuscript is interesting and well written. The authors give adequate description of the methodological approach. The experimental data are in line with the scope of work. The aim of research fits with the topics covered by the magazine.

I only suggest the authors to indicate the extraction yields and the amount of the purified compounds.

Minor points:

Lines 64 and 252: add references

Line 125: remove dot after C12H14Br2N2O2

Line 147: add space after C12H15Br2N2O

Lines 315, 320 and 325: remove space between “0.1” and “%”

Lines 346 and 348: replace “For” with “for”

Lines 372 and 374: replace “was” with “

Lines 382 and 395: add space between “37” and “°C”

Line 392: replace “compounds concentration” with “compound concentrations”

Line 395: add space between “1” and “h” and between “37” and “°C”

Lines 410 and 407: add spectrophotometer model

Line 405: the subject is missing

Figure 6: axis label (% LPS) is not clear. The authors stimulates cells with lipopolysaccharide (LPS) at 500 ng/mL concentration!!

Author Response

I only suggest the authors to indicate the extraction yields and the amount of the purified compounds.

Information added for the known compounds

Lines 64 and 252: add references

One reference was added

Line 125: remove dot after C12H14Br2N2O2

Done

Line 147: add space after C12H15Br2N2O

Done

Lines 315, 320 and 325: remove space between “0.1” and “%”

Done

Lines 346 and 348: replace “For” with “for”

Done

Lines 372 and 374: replace “was” with “

Done

Lines 382 and 395: add space between “37” and “°C”

Done

Line 392: replace “compounds concentration” with “compound concentrations”

Done

Line 395: add space between “1” and “h” and between “37” and “°C”

Done

Lines 410 and 407: add spectrophotometer model

Information added

Line 405: the subject is missing

Corrected

Figure 6: axis label (% LPS) is not clear. The authors stimulates cells with lipopolysaccharide (LPS) at 500 ng/mL concentration!!

It is indeed % of the control.

Reviewer 2 Report

Bromotryptamine and bromotyramine derivatives from the Tropical Southwestern Pacific sponge Narrabeena nigra, by Gordo et al.

The manuscript outlines the isolation of 18 brominated alkaloids, 8 new and 10 known, from the sponge N. nigra. The new compounds were structurally elucidated and tested for activity in two different bioassays. The results show that nine of the compounds have interesting neuroprotective activity.

The manuscript is in general concise, but there are some issues with English grammar and writing (however, the quality improved with the second submission of the manuscript).

General comments:

The structures of the known compounds should also be included in a figure. This would make it easier for the reader to read the manuscript and compare the structures.

The authors state in the ‘results&discussion’ (L72-73) that they purified 18 compounds from some of the fractions. How did they identify the compounds to isolate? Did they analyse the fractions using HR-MS, or did they analyse the crude extract? 

The authors have isolated 18 compounds, and the bioactivity of all compounds were tested. However, only information regarding the isolation of the new compounds (i.e. 1-8) is given, even though analytical information (i.e. NMR and MS) for all compounds is given in SI. The authors should provide some information on how the compounds 9-18 were isolated, and what amounts of pure compounds they were able to isolate.

No information on the method applied for the prediction of metabolites and their networking is given.

Detailed comments is given in a separate file.

Author Response

The structures of the known compounds should also be included in a figure. This would make it easier for the reader to read the manuscript and compare the structures.

The structures of all compounds are all drawn in the figure of the metabolic pathway.

The authors state in the ‘results&discussion’ (L72-73) that they purified 18 compounds from some of the fractions. How did they identify the compounds to isolate? Did they analyse the fractions using HR-MS, or did they analyse the crude extract?

In fact we isolated and characterized all the compounds we could from the sponge specimen as long as we had enough quantity for NMR studies.

The authors have isolated 18 compounds, and the bioactivity of all compounds were tested. However, only information regarding the isolation of the new compounds (i.e. 1-8) is given, even though analytical information (i.e. NMR and MS) for all compounds is given in SI. The authors should provide some information on how the compounds 9-18 were isolated, and what amounts of pure compounds they were able to isolate.

Information is given for all isolated compounds new or known in the material and method and references are given for the reference article.

No information on the method applied for the prediction of metabolites and their networking is given.

Information has been added sorry for the mistake.

Detailed comments is given in a separate file.

Most comments were addressed

Reviewer 3 Report

Manuscript Number: marinedrugs-504945

Author(s): Olivier Thomas et al.

Title: “Bromotryptamine and bromotyramine derivatives from the Tropical Southwestern Pacific sponge Narrabeena nigra "

This paper deals with the isolation and structural characterization of seven new bromotryptamine and one new bromotyramine together with 10 known compounds from sponge Narrabeena nigra. Their structures were elucidated from extensive NMR and HRMS studies while the absolute configuration of two of them were determined by comparison between the experimental and calculated ECD spectra. The biological analysis of all the isolated compounds displayed that some of them exhibit neuroprotective activity by reducing oxidative damage and did not show almost no toxic effect up to 10 µM. Furthermore, a biosynthetic hypothesis is displayed and some possible biosynthetic intermediates were proposed using the webserver MetWork.

The subject matter is appropriate for Marine Drugs. The work is of sufficient quality and quantity to be published after some minor revisions.

Important question

Is this research the first chemical study of specimens of the genus Narrabeena? If this is true, authors must specify this fact in the manuscript.

Minor corrections,

Page 3, lines 93: As the fragments “m/z 344.9 and 301.9” were obtained by HR/MS, the four-digit decimals must be shown. The same in Page 3, line 106, etc.

Page 4, line 129: Correct “an unique”

Page 10, line 283: The text in the figure 7 is illegible.

Page 11, line 337: Insert space in “furnished1”

Page 12, line 376: bioassays must be written as Bioassays”

References

Issue numbers of the references must be deleted

Organism’s names must be written in italic e.g Smenospongia in ref 12, Aplysina and smenospongia in ref 17, etc.

Page 15, line 498: smenospongia must be written as Smenospongia

Page 15, line 500: Insert spaces in “PolyfibrospongiaMaynardii”

Author Response

Important question

Is this research the first chemical study of specimens of the genus Narrabeena? If this is true, authors must specify this fact in the manuscript.

We highlighted this point in the conclusion.

Minor corrections,

Page 3, lines 93: As the fragments “m/z 344.9 and 301.9” were obtained by HR/MS, the four-digit decimals must be shown. The same in Page 3, line 106, etc.

Corrections made

Page 4, line 129: Correct “an unique”

Not changed

Page 10, line 283: The text in the figure 7 is illegible.

We increased the resolution but the software does not allow better

Page 11, line 337: Insert space in “furnished1”

Done

Page 12, line 376: bioassays must be written as Bioassays”

Done

References

Issue numbers of the references must be deleted

Organism’s names must be written in italic e.g Smenospongia in ref 12, Aplysina and smenospongia in ref 17, etc.

Page 15, line 498: smenospongia must be written as Smenospongia

Page 15, line 500: Insert spaces in “PolyfibrospongiaMaynardii”

Corrections made